# Strategies for Improving the CO_2_ Adsorption Process of CPO-27-Mg through Thermal Treatment and Urea Functionalization

**DOI:** 10.3390/ma16010117

**Published:** 2022-12-22

**Authors:** Agustín A. Godoy, Dimar Villarroel-Rocha, José Joaquín Arroyo-Gómez, Celeste Bernini, Griselda Narda, Karim Sapag

**Affiliations:** 1Instituto de Investigaciones en Tecnología Química (INTEQUI-CONICET), Universidad Nacional de San Luis, Almirante Brown 1455, San Luis 5700, Argentina; 2Instituto de Nanociencia y Nanotecnología (INN-CONICET), Centro Atómico Constituyentes, Comisión Nacional de Energía Atómica (CNEA), Avenida General Paz 1499, San Martín, Buenos Aires 1650, Argentina; 3Laboratorio de Solidos Porosos (LabSoP), Instituto de Física Aplicada (INFAP-CONICET), Universidad Nacional de San Luis, Avenida Ejercito de los Andes 950, San Luis 5700, Argentina; 4Departamento de Almacenamiento de la Energía (DADLE), Subgerencia Operativa de Energía y Movilidad, Instituto Nacional de Tecnología Industrial (INTI), Avenida General Paz 5445, San Martín, Buenos Aires 1650, Argentina; 5Consejo Nacional de Investigaciones Científicas y Técnicas (CONICET), Godoy Cruz 2290, Argentina

**Keywords:** MOFs, CPO-27, functionalization, CO_2_ adsorption

## Abstract

In this work, the influence of degassing temperature and urea functionalization were investigated as ways to improve the CO_2_ adsorption performance of CPO-27-Mg. Through post-synthesis modification treatments, four samples with different degrees of urea functionalization were obtained, incorporating 10, 25, 50, and 100% of urea concerning the metal sites of the MOF. Alternatively, the influence of the degassing temperature of the non-functionalized MOF between 70 and 340 °C was also evaluated. The resulting compounds were characterized by N_2_ adsorption–desorption isotherms at −196 °C using TGA-MS, FTIR, and PXRD. Finally, the thermally treated and functionalized CPO-27-Mg was evaluated for CO_2_ capture.

## 1. Introduction

In recent decades, an unusual elevation in the average planetary temperature has occurred due to an increase in greenhouse gases. In this sense, the danger of imminent excessive global warming has alerted the world population due to its harmful environmental, social, economic, and health effects [1]. Currently, the largest source of greenhouse gas emissions of anthropogenic origin is from burning fossil fuels such as coal, natural gas, and oil to produce electricity and heat [2]; as a consequence, an increase in the annual average global CO_2_ concentration of more than 40% has taken place since the beginning of the industrial revolution, i.e., from 280 ppm in the mid-18th century to 415.58 ppm in July of 2022, according to the latest U.S. National Oceanic and Atmospheric Administration (NOAA) measurement. If this trend maintains its growth rate, it is estimated that the concentration will exceed 500 ppm by 2050 [3]. Due to its physicochemical characteristics, abundance, and indirect effects, CO_2_ has the most significant influence on the greenhouse effect; therefore, the development of mitigation technologies, known as carbon capture and storage (CCS), have emerged as a promising alternative to try to regulate its concentration [4].

Different CCS techniques have been applied in post-combustion CO_2_ generation conditions, where CO_2_ is separated from a low-pressure gas stream [5]. Commonly, the use of alkanolamine aqueous solutions is efficient in capturing CO_2_ from industrial streams [6]. However, the high cost involved in the regeneration process of this sorbent limits its application on a large scale, leading to the development of solid CO_2_ sorbents such as activated carbons, zeolites, and porous polymers, with significantly lower regeneration costs than traditional amine scrubbers [7]. Within these options, metal–organic frameworks (MOFs)—structures with permanent porosity conformed by metallic clusters and organic ligands—have emerged as candidates with enormous potential to be applied in gas capture processes [8,9]. This is not only due to their high specific surface areas, which have considerably exceeded the capabilities of traditional materials, but also for special characteristics related to their structural versatility; it is possible to tailor their functionality by introducing polar functional groups that have a strong interaction with CO_2_ at the pore surfaces. In this way, the adsorbate–adsorbent affinities in MOFs can be controlled and optimized. 

The activation process is a crucial factor in MOF performance of [10,11,12], i.e., removing guest molecules from the porous structure and maintaining its structural integrity to obtain a high degree of porosity. Different strategies have been implemented for this, such as heat and vacuum treatment, solvent exchange, supercritical CO_2_ (scCO_2_) exchange; freeze drying; and chemical treatment [12,13,14]. Numerous reports have shown that, due to the simplicity and efficiency in evacuation of the host species, a combination of thermal treatments and solvent exchange is one of the best activation strategies, where the MOF is heated below its decomposition temperature.

For solvent exchange, the selected solvent should have a lower boiling point (e.g., CHCl_3_, CH_3_OH) than that used in the MOF synthesis (e.g., DMF), making it easily removable under a low vacuum. For example, Kaye et al. report a heat treatment of Zn_4_O(BDC)_3_ (MOF-5, BDC^2−^ = 1,4-benzenedicarboxylate), where the guest molecules are evacuated to leave a rigid metallic organic scaffold stable up to 400 °C under vacuum [15], and Alezi et al. report that Al-soc-MOF structures preserved their optimal porosity after heating up to 340 °C under vacuum [16]. As a result, the removal of guest species generates coordinatively unsaturated metal sites (CUSs), which can act as Lewis acids and interact with CO_2_.

CPO-27-Mg (also known as Mg-MOF-74 [17] or Mg/DOBDC [18]), a cylindrical pore-structured MOF, holds the record for the highest gravimetric adsorption of CO_2_ (i.e., 8.6 mmol g^−1^) at 25 °C and 1 bar [19]. This behavior can be associated with the large density of magnesium CUSs and the influence of Mg-O, which improves CO_2_ attraction/affinity. 

Once MOF channels are evacuated, the introduction of highly CO_2_-philic moieties into the pore surface is an effective way to improve CO_2_ adsorption capacity. In addition, post-synthetic modification (PSM) by impregnation with amines can also enhance CO_2_ adsorption capacity due to the high amine−CO_2_ reaction equilibrium constants. Particularly, functionalization of CPO-27-Mg with multi-amino species like ethylenediamine (EDA) [20,21,22] and tetraethylenepentamine (TEPA) [23] have been shown to provide not only enhanced CO_2_ binding under dilute and moderate pressure intervals but also improved the overall stability when compared with the unmodified MOF.

In this work, the optimization of CPO-27-Mg as a CO_2_ adsorbent was proposed, following two strategies: through thermal treatments at different temperatures on the non-functionalized MOF and by PSM through impregnation with urea, whose functional amino groups are presented as species with high affinity for CO_2._ Urea was selected to perform impregnation experiments on CPO-27-Mg because its small size would make it possible to incorporate into the MOF micropores. Furthermore, its chemical structure comprises a carbonyl group that can efficiently interact with the Mg-CUSs, leaving two free amino groups with the possibility of acting as Lewis bases by sharing their isolated pair of electrons with CO_2_. 

## 2. Materials and Methods

### 2.1. Reagents

We obtained magnesium nitrate hexahydrate (Mg(NO_3_)_2_·6H_2_O) (98%, Sigma-Aldrich, St. Louis, MO, USA), 2,5-dihydroxyterephthalic acid (DHTA) (98%, Sigma-Aldrich), N,N-dimethylformamide (DMF) (99.8%, Anedra, Buenos Aires, Argentina), ethanol (99.8%, Dorwil, Buenos Aires, Argentina), methanol (99.8%, Anedra, Buenos Aires, Argentina), urea, and isopropyl alcohol (99.8%, Dorwil). All reagents were used as received without further purification.

### 2.2. CPO-27-Mg Synthesis

The metal–organic framework was obtained following the procedure reported elsewhere [24], i.e., 457 mg of Mg(NO_3_)_2_·6H_2_O (1.83 mmol) and 107 mg of DHTA (0.53 mmol) were dissolved in a mixture of solvents composed of 44 mL of DMF, 3 mL of ethanol, and 3 mL of distilled water. First, the solution was heated at 125 °C for 21 h in a digestion bomb (60 mL of internal volume). After that, the reactor was cooled to room temperature, the solvent was decanted, and the remaining solid was put in contact with DMF (30 mL) for two days, replacing the solvent daily. After this time, the solvent was exchanged for methanol (30 mL), replacing it daily for three days. At the end of this process, the solid was dried under a vacuum at room temperature.

### 2.3. CPO-27-Mg Post-Synthetic Modification with Urea-Isopropyl Alcohol Solutions

The post-synthesis procedure involved a thermal treatment under dynamic vacuum conditions as a first step, employing the synthesized CPO-27-Mg sample. The thermal treatment was carried out by applying a heating rate of 1 °C.min^−1^, from room temperature to ca. 230 °C, and then keeping this temperature for six hours. The cooling rate was fixed at 5 °C.min^−1^ until the system returned to room temperature. The solid was kept in a glass vessel under dynamic vacuum throughout the entire thermal process. This procedure was performed to remove the solvent (coordinated and non-coordinated) located inside the framework pores. In a second step, the resulting activated samples were put in contact with urea solutions in isopropyl alcohol to obtain materials with four different urea-grafted contents. The suspensions were stirred for 24 h under reflux at 65 °C, and after that the solid was separated by centrifugation. The samples obtained by this procedure were dried under vacuum at room temperature.

### 2.4. Powder X-ray Diffraction (PXRD)

PXRD patterns were obtained with a Rigaku Ultima IV diffractometer (Rigaku Corporation, Tokyo, Japan), using Cu Kα radiation, 1.5418 Å, and quartz as an external calibration standard. The best counting statistics were achieved using a continuous mode, between 4–40 ° in 2θ, with a scanning rate of 2° min^−1^ and a step of 0.03°.

### 2.5. Fourier-Transform Infrared Spectroscopy (FTIR)

FTIR spectra were recorded with a Nicolet Protégé 460 spectrometer in the 4000–400 cm^−1^ range with 64 scans and a spectral resolution of 4 cm^−1^ using the KBr pellet technique.

### 2.6. Thermogravimetric Analysis (TGA)–Mass Spectrometry (MS)

Thermogravimetric analysis (TGA) was carried out using an SDT Q600 thermal analyzer (T.A. Instruments, New Castle, DE, USA). The samples (ca. 4 mg) were placed in an alumina pan and then heated from room temperature up to 600 °C with a heating rate of 5 °C.min^−1^ under a He atmosphere at 50 mL.min^−1^. The evolved gases were analyzed using a Discovery mass spectrometer, using the multiple ion detection mode. The obtained data were analyzed using Universal Analysis 2000 software from T.A. Instruments.

### 2.7. Gas Adsorption

N_2_ adsorption–desorption isotherms were measured at −196 °C in an ASAP 2000 sorption analyzer from Micromeritics, and CO_2_ adsorption isotherms were measured at 0 °C until a pressure of 10 bar in an ASAP 2050 sorption analyzer from Micromeritics. The samples were degassed between 70–340 °C in a vacuum for 8 h. The apparent specific surface area (S_BET_) was calculated by the BET equation [25] using the N_2_ adsorption data and following the Rouquerol criteria [26]. The micropore volumes (V_μP-N2_) were obtained by applying the Dubinin–Radushkevich equation [27] using the N_2_ adsorption data. The total pore volume (V_TP_) was calculated using the Gurvich rule at p/p^0^ = 0.98 from N_2_ adsorption data [28]. The pore size distributions (PSDs) were evaluated by the modified Horvath–Kawazoe method for pores with cylindrical geometries [29,30] using CO_2_ adsorption data. CO_2_ adsorption performance at 25 °C for all materials was studied in an ASAP 2050 automatic manometric sorptometer from Micromeritics up to a pressure of 10 bar. Before the measurements, the materials were degassed between 70– 340 °C for 12 h under a vacuum.

All non-functionalized and functionalized samples were employed in adsorption experiments and characterizations within 24 h of their preparation. During this time, all samples were kept under ambient conditions.

## 3. Results and Discussion

### 3.1. Characterization

#### 3.1.1. DRXP and FTIR

The materials studied in this work include the CPO-27-Mg MOF in its bare form and functionalized with urea. Through an impregnation treatment in urea-isopropyl alcohol solutions, samples with 10, 25, 50, and 100 mmol% urea were obtained. The percentage of mmol% urea incorporated in the post-synthesis procedure indicated the relative amount of coordinatively unsaturated sites (CUSs) that could be functionalized. These samples were named CPO-27-Mg-F10, F25, F50, and F100, respectively.

The characterization of all materials started with verifying their crystalline nature using powder X-ray diffraction (PXRD). Figure 1 shows the diffraction patterns obtained compared with the simulated powder pattern of CPO-27-Mg. As can be seen, in all cases, CPO-27-Mg was obtained as the only crystalline phase without impurities or secondary phases, which can be evidenced by the agreement with the reflections of the simulated powder pattern and the absence of additional reflections. Additionally, the functionalized samples show a decrease in crystallinity due to the post-synthesis impregnation treatments. In this series, the presence of the two main reflections stands out, i.e., those located at 6.9° and 11.9° of 2θ, corresponding to the crystallographic planes (2 1 0) and (3 0 0), respectively. As expected, the treatment decreased the crystallinity of the MOF, and this fact could be analyzed by determining the average crystallite size (Dp) through the Scherrer equation (Equation (1)) [31]:Dp = (0.94 × λ)/(β × cosθ) (1)
where Dp = average crystallite size, β = line broadening in radians, and θ = Bragg angle, λ = X-ray wavelength.

For this analysis, only the two main reflections were considered. The Dp of non-functionalized CPO-27-Mg was taken as a reference point for this comparative analysis, with a calculated value of ~54.8 nm, while those of the functionalized samples were 40.2, 40.0, 50.8, and 45.7 nm for the F10, F25, F50, and F100 samples, respectively. These results show that the Dp decreased by between 7 and 27% after the post-synthesis procedure, in relation to the as-synthesized CPO-27-Mg. Table 1 shows the results obtained.

Figure 1 allows a verification of a correlation between the decrease in Dp and the reflection intensities for F10 and F25, which are significantly lower than in the F50 and F100 samples.

The study of urea incorporation in the MOF began with a qualitative analysis by FTIR spectroscopy focused on the spectral zones, giving more information about the functional groups of the urea molecule. In agreement with previous works, CPO-27-Mg shows the characteristic bands of this material [22,24]. On the other hand, the urea bands were assigned based on data reported in the literature [32,33]. As shown in Figure 2, the broad band between 3500–3300 cm^−1^, corresponding to the vibrational stretching (ν_as_ and ν_s_) of the O-H groups of the hydration water molecules, is centered at ca. 3380 cm^−1^ for the bare CPO-27-Mg. In some of the functionalized samples is possible to observe the splitting of this band, with two components located at 3437 cm^−1^ and 3351 cm^−1^. These bands are clearly notable in F50 and F100 and can be assigned to the vibrational stretching of the N-H groups of urea. Additionally, the stretching of the urea C=O group, located at 1681 and 1628 cm^−1^ in the pure urea spectrum, is observed in samples F50 and F100, while they less noticeable in the spectra of the samples with a lower degree of functionalization. A similar behavior is observed when the urea bands located at 1462 cm^−1^ and 1158 cm^−1^, attributed to the bending modes (δ N-H) of amino groups, are looked for on the functionalized sample spectra. An interesting result is a shift observed in the signal of the urea carbonyl group from 1681 cm^−1^ for pure urea to 1666 cm^−1^ in compounds F50 and F100. This fact could be related to the interaction between the carbonyl group and the activated MOF Mg centers. 

#### 3.1.2. TGA-MS

Thermogravimetric analysis allowed evaluation of the thermal stability and temperatures where the mass decrease events occur in both functionalized and non-functionalized MOFs to determine the optimal temperature ranges for the degassing process.

The TGA curve and schematic representation of the stages of the activation process of non-functionalized CPO-27-Mg are presented in Figure 3. The blue box represents the initial state, where the non-coordinated solvent molecules start to leave the pore framework; the green box denotes the following state, where coordinated solvent remains inside the MOF pore; and the orange box represents the situation achieved when the MOF is fully activated because all coordinated solvent has left the pore. Additionally, Figure 4 shows the mass/charge signals associated with water (*m*/*z* 18), methanol (*m*/*z* 29 and 31), and DMF (*m*/*z* 73) molecules; this allows us to associate the weight changes to the release of the species that fill the MOF channels (see MS of the solvents in Appendix A). The TGA behavior of the CPO-27-Mg synthesized here is similar to that reported elsewhere [18]. 

Figure 3 shows that the loss of the solvent occurs approximately between 50 and 290 °C, with a weight loss of 27.6 %. The *m*/*z* 18 water signal is observed mainly in the range of 50–125 °C with two intensity maxima, one centered at 60 °C and the other at 100 °C, which can be attributed to hydration molecules; DTG corroborates the two maxima. On the other hand, the signals *m*/*z* 29 and 31, characteristic of methanol, appear in a wide range of temperatures between 50 and 250 °C, showing the same intensity profile. The first maximum is around 68 °C, close to the boiling point of methanol (65 °C), which is attributed to non-coordinated species. The following two bands, one at 103 °C and the largest at 159 °C, are attributed to coordinated methanol. Finally, the *m*/*z* 73 signal, characteristic of DMF, was not observed in the range of temperatures evaluated, indicating that the solvent exchange process completely removed the DMF.

It can be stated that the solvent is entirely released above ca. 290 °C, because no significant decrease in mass is observed and the *m*/*z* signals of water and methanol are almost negligible. Still, there is a range of temperatures in which it is possible to continue adjusting the activation efficiency of the material before reaching its degradation point (ca. 360 °C). For this purpose, the evolution of the fragment *m*/*z* = 44 (CO_2_) was carefully evaluated; this fragment indicates the production of CO_2_ as an oxidation product of the organic components of the MOF. An abrupt increase in the *m*/*z* 44 signals is evident from 350 °C, with a maximum intensity at 393 °C. This signal is consistent with the decrease in mass observed in the TGA and with the maximum in the derivative signal at 400 °C. Based on this result, the temperature range of 250–330 °C (orange shaded section in Figure 4) was selected to analyze the influence of temperature on the degassing process in CPO-27-Mg. The usual degassing temperature for this MOF is 250 °C [22,24,34].

TGA-MS analysis was also performed on the functionalized compounds (see Appendix A), evaluating the signals of the main fragments of urea: *m*/*z* 17, 44, and 60 (see M.S. of urea in Appendix A). Figure 5a shows the TGA for pure urea. As shown by the derivative signal, urea begins to degrade from 134 °C, reaching a maximum degradation rate at around 167 °C. Figure 5b shows that the signals *m*/*z* 17 and 44 are both observed in the range of 164–210 °C until reaching a maximum at ca. 160 °C.

For the functionalized compounds, the general trend indicates that *m*/*z* 17 and 44 signals increase according to the degree of functionalization, being noticeable in F50 and F100, but negligible in F10 and F25. As for *m*/*z* 17 (see Figure 6a), it is possible to observe in all cases a band below 100 °C, which is attributed to the HO^+^ species coming from hydration water molecules. In contrast, the band located in the 200–260 °C range can be attributed to the NH_3_^+^ fragment of urea, which appears at a higher temperature than in pure urea (134–210 °C), evidencing a confinement effect for the framework pore. On the other hand, *m*/*z* 44 (see Figure 6b) presents maxima in its signals at 255 °C and 234 °C for F50 and F100, respectively, while in the samples with lower degrees of functionalization, only small signals can be observed around 250–275 °C. It is interesting to note that both signals (*m*/*z* 17 and 44) occur at higher temperatures than the corresponding ones in pure urea, which can be explained in terms of the existence of an interaction between this species and the MOF pore walls.

As for the non-functionalized CPO-27-Mg, based on the results of the TGA-MS analysis, the temperature range to be evaluated during the subsequent degassing process was determined, with below 200 °C appearing as suitable. At that point, the signals *m*/*z* 17 and 44 show high intensities.

### 3.2. Effect of the Degassing Conditions on CO_2_ Adsorption

CO_2_ adsorption isotherms at 25 °C for CPO-27-Mg degassed between 140 and 340 °C are displayed in Figure 7. The adsorption capacities of CPO-27-Mg reached values of 2.7 and 8 mmol g^−1^ at 1 bar (Figure 7a) and 4.6 and 12.4 mmol g^−1^ at 10 bar (Figure 7b), respectively. The yields at different pressures showed an increasing trend as the degassing temperature increased; the compound CPO-27-Mg degassed at 330 °C (CPO-27-Mg Tdegas330) exhibited the best CO_2_ adsorption capacity of 8 mmol g^−1^ at 1 bar and 12.4 mmol g^−1^ at 10 bar, increasing up to 170% when compared to CPO-27-Mg degassed at 140 °C (CPO-27-Mg Tdegas140). The adsorption capacity of the degassed compound at 340 °C was the same as that degassed at 330 °C.

Higher degassing temperatures were not tested because of sample degradation, as evidenced by the TGA-MS curves of CPO-27-Mg (Figure 3 and Figure 4).

The effect on the textural properties of the samples under different degassing temperatures was analyzed using N_2_ adsorption–desorption isotherms at −196 °C. The degassed CPO-27-Mg compounds at 140, 250, and 330 °C were selected, and their corresponding adsorption–desorption N_2_ isotherms are displayed in Figure 8a. According to IUPAC classification, all samples present an isotherm Type I(b) at low relative pressure, with an H4 hysteresis loop, characteristic of microporous materials having supermicropores [26]. Finally, the adsorption–desorption isotherms of N_2_ show the same trend as CO_2_, i.e., increasing in adsorbed quantity as the degassing temperature increases.

The textural properties of the CPO-27-Mg compounds degassing at different temperatures are shown in Table 2, obtained from N_2_ and CO_2_ isotherms.

It is noticeable that the increase in micropore volumes and specific surface area are associated with the degassed temperature increase. In addition, almost the same micropore volume values are obtained for N_2_ and CO_2_, which is correct because it seems that these samples have only supermicropores (pore size between 0.7 and 2 nm), which are the ones most easily detected with both adsorbates. The modal micropore size of 1.3 nm shown in the PSD of Figure 7b, obtained with CO_2_ adsorption data at 0 °C, is in agreement with the results reported by other authors [18,35]. In general, the textural properties of the composites show variations expected concerning those in previous studies with CO_2._ The degassing temperature increased the supermicropore volume where the CO_2_ adsorption was cumulated.

### 3.3. Urea Functionalization

In this section, the most convenient degassing temperatures were chosen following the TGA-MS analysis, looking for the optimum temperature at which the uncoordinated urea and the residual solvent molecules adsorbed in the framework pores can be evacuated without removing the coordinated urea.

For the first experiment, 140 °C was chosen, i.e., where the hydration water and coordinated methanol have left the pore framework (Figure 6a), and Figure 9 shows the CO_2_ adsorption at 25 °C of the degassed functionalized compounds at this temperature. The CO_2_ adsorption capacities reached 1.2 and 5.6 mmol g^−1^ at 1 bar (see Figure 9a) and 2.6 and 8.9 mmol g^−1^ at 10 bar (see Figure 9b). Compounds representing 50 and 100% functionalization show a low CO_2_ adsorption capacity, indicating that the pores could remain blocked. 

To determine whether other degassing temperatures could lead to better adsorption capacities in functionalized samples, lower and higher degassing temperatures were studied. To examine the degradation temperature of coordinated urea (see Figure 6b), temperatures of 70 and 190 °C were selected. These studies were performed by choosing the best compounds (F10 and F25, for their adsorption capacity) from the previous study (Figure 9).

Figure 10 shows the CO_2_ adsorption at 25 °C of the functionalized compounds at 10 and 25% degassed at 70 and 190 °C. The CO_2_ adsorption capacities reached values of 1.9 and 9 mmol g^−1^ at 1 bar (Figure 10a) and 4 and 15.2 mmol g^−1^ at 10 bar (Figure 10b), respectively.

According to the TGA analysis, urea decomposition begins above 100 °C (Figure 5a); then, at 70 °C degassing temperature, it is probable that the urea is still blocking the available active sites for CO_2_ adsorption (ultramicropores). It does not occur at 190 °C, where the highest adsorption capacity appears, related to the almost full release of the functionalized compound from the channels retaining a proportion of coordinated urea that improved the CO_2_ adsorption capacity.

The compound CPO-27-Mg-F25 degassed at 190 °C (CPO-Mg-F25 Tdegas190) exhibited the best adsorption capacity of 9 mmol g^−1^ at 1 bar and 15.2 mmol g^−1^ at 10 bar. These results allow identification of the best combination of initial urea amount and optimum degassing treatment to improve the adsorption capacity of this MOF.

Figure 11 compares the CO_2_ adsorption isotherms at 25 °C with the best performance achieved by both optimization methods up to 1 bar (see Figure 11a) and 10 bar (see Figure 11b). The improvement reached by the urea functionalization strategy leading to the CPO-27-Mg-F25 sample and the degassing treatment at 190 °C led to an outstanding CO_2_ adsorption capacity that exceeds that of CPO-27-Mg degassed at 330 °C by 12.36 and 24.55% at 1 bar and 10 bar, respectively.

To compare the CO_2_ capture performance of the prepared materials, Table 3 summarizes the CO_2_ adsorption capacities of other functionalized MOFs, including CPO-27-Mg, and other porous materials. From this table, it is possible to appreciate the excellent status of the prepared samples.

## 4. Conclusions

Optimization strategies for the CO_2_ adsorption capacity of the MOF CPO-27-Mg were investigated through degassing temperature on the raw material and urea functionalization. The results obtained using TGA-MS determined the optimal temperature ranges for descaling of the materials. The CO_2_ isotherms at 25 °C at 1 bar and 10 bar of the non-functionalized CPO-27-Mg were evaluated in the range of 140–340 °C, obtaining the best performance at 330 °C for both pressures, with adsorption capacities of 8.01 and 12.26 mmolg^−1^ respectively. The best result in this study was obtained for a material prepared with 25% urea functionalization (CPO-27-Mg-F25), reaching CO_2_ adsorption values of 9 and 15.26 mmolg^−1^ at 1 and 10 bar, respectively, at a degassing temperature of 190 °C. In this way, the methodologies performed in this work show how the combination of two strategies leads to optimization of the CO_2_ adsorption capacity of a well-known MOF material. The determination of thermal treatments that ensure the complete emptying process of the microporous channels along with the post-synthesis modification (PSM) by introducing highly CO_2_-philic species on the pore surface of the MOF results in an effective way to improve its adsorption capacity.

## Figures and Tables

**Figure 1 materials-16-00117-f001:**
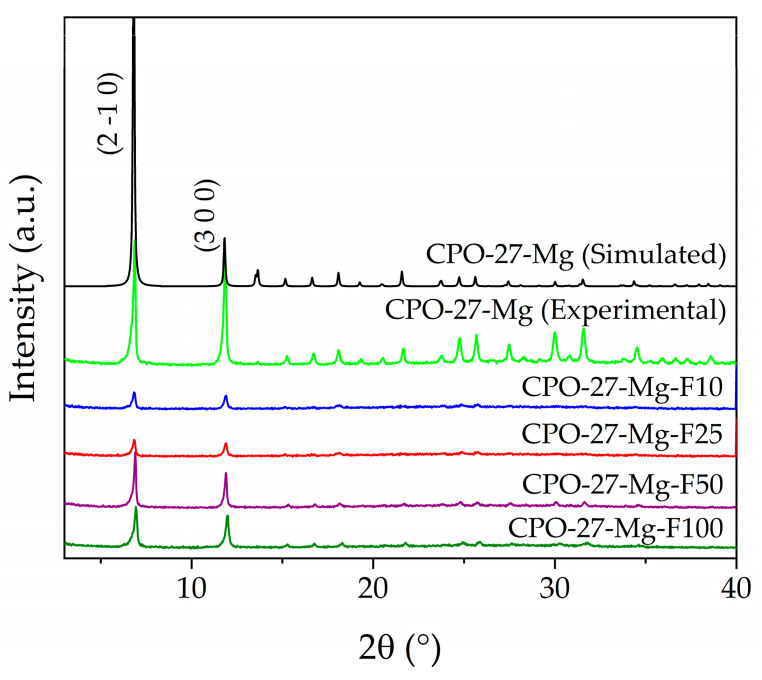
X-ray powder diffraction of functionalized and non-functionalized compounds.

**Figure 2 materials-16-00117-f002:**
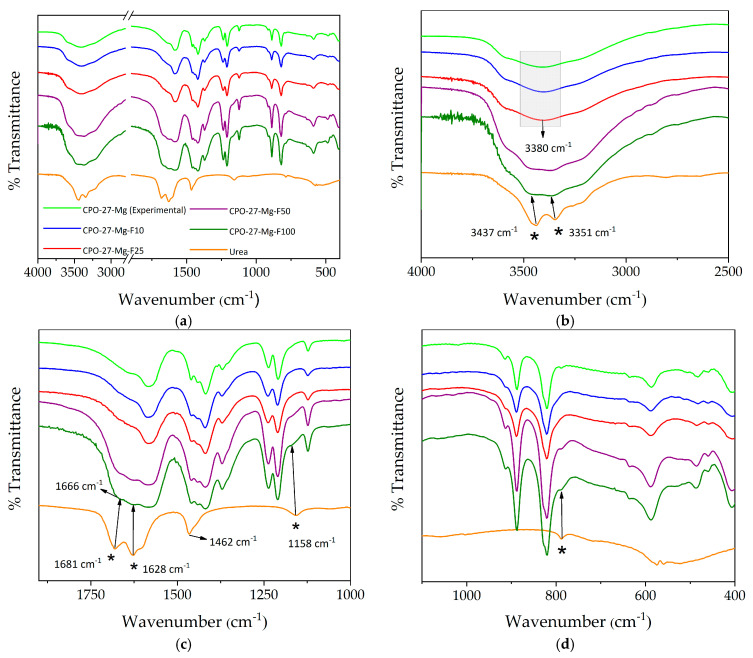
(**a**–**c**) FTIR spectra of the synthesized and functionalized compounds compared to the spectrum of urea in the range of 4000–400 cm^−1^. (**b**–**d**) show the magnification of different zones of the spectra (bands assigned to urea are marked with *).

**Figure 3 materials-16-00117-f003:**
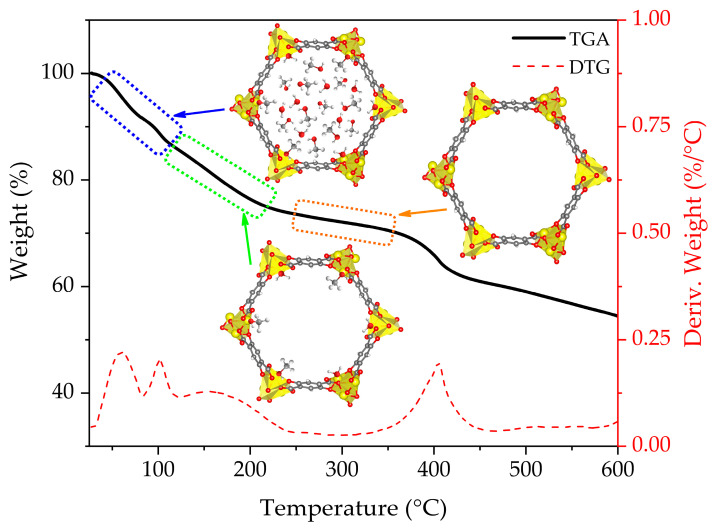
TGA-DTG curves for CPO-27-Mg.

**Figure 4 materials-16-00117-f004:**
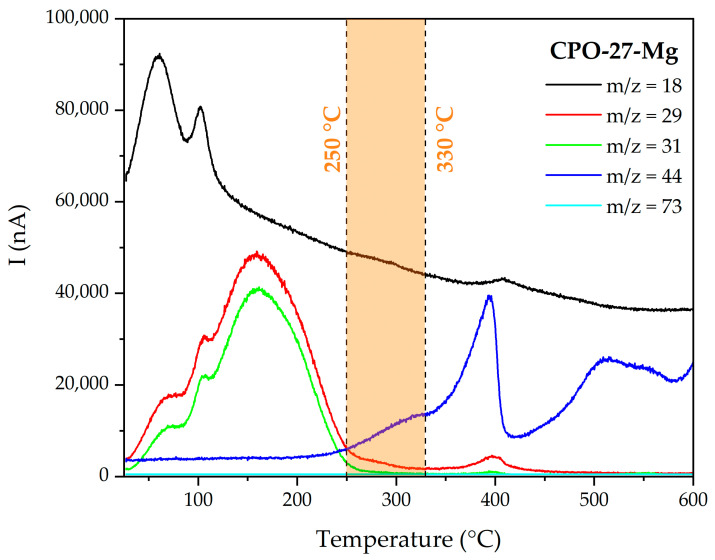
MS multiple ion detection curves for CPO-27-Mg.

**Figure 5 materials-16-00117-f005:**
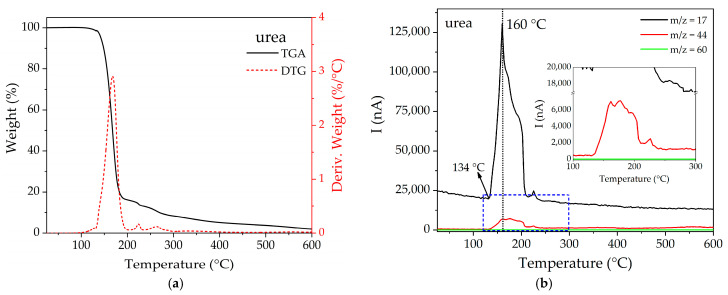
(**a**) TGA and (**b**) MS multiple ion detection curves for pure urea.

**Figure 6 materials-16-00117-f006:**
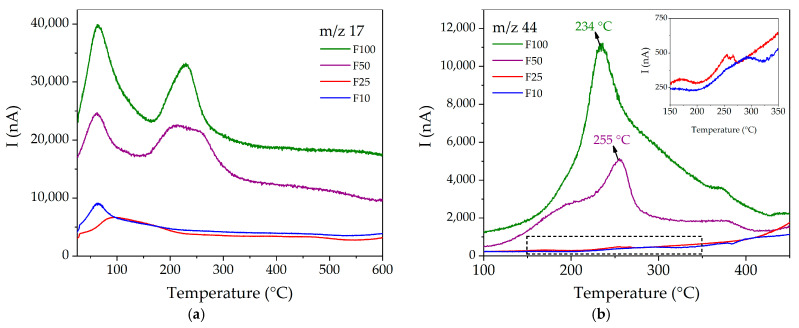
MS multiple ion detection curves for (**a**) *m*/*z* = 17 and (**b**) 44 for the functionalized compounds.

**Figure 7 materials-16-00117-f007:**
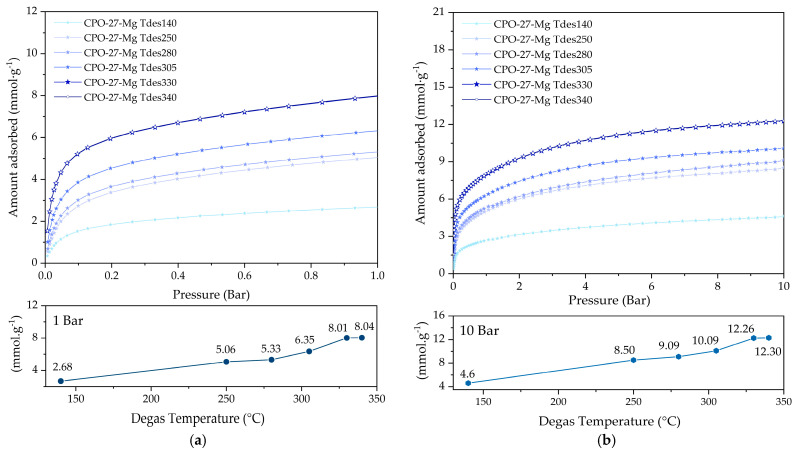
CO_2_ adsorption at 25 °C of CPO-27-Mg compounds up to (**a**) 1 bar and (**b**) 10 bar of pressure.

**Figure 8 materials-16-00117-f008:**
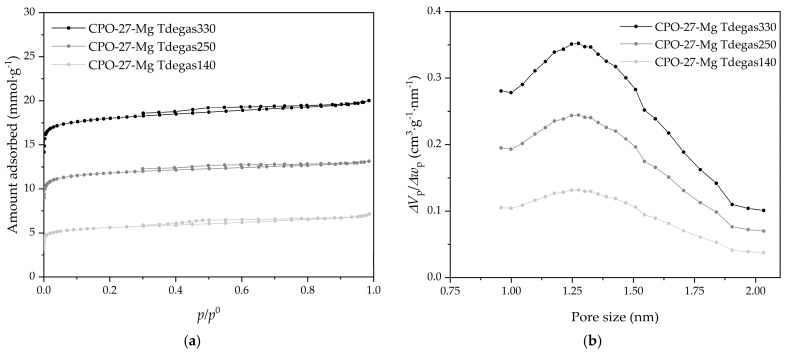
(**a**) N_2_ adsorption–desorption isotherms at −196 °C and (**b**) pore size distributions (obtained with CO_2_ adsorption data) of the CPO-27-Mg compounds.

**Figure 9 materials-16-00117-f009:**
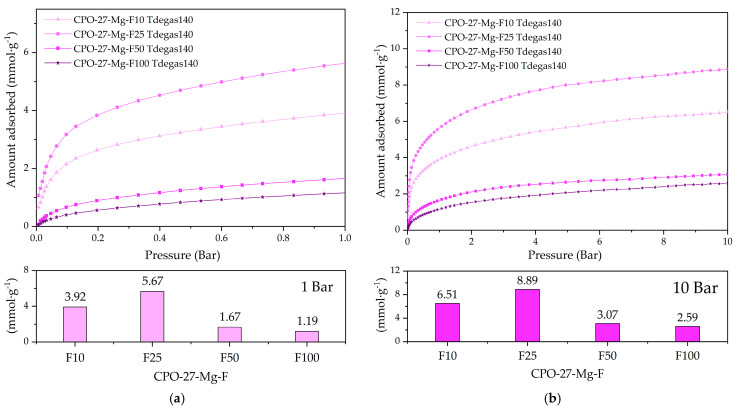
CO_2_ adsorption at 25 °C of the degassed functionalized compounds at 140 °C up to (**a**) 1 bar and (**b**) 10 bar of pressure.

**Figure 10 materials-16-00117-f010:**
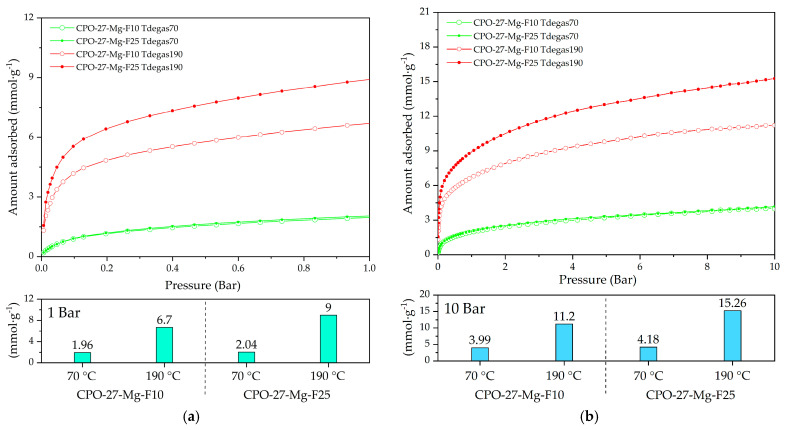
CO_2_ adsorption at 25 °C of CPO-27-Mg-F10 and CPO-27-Mg-F25 degassed at 70 and 140 °C up to (**a**) 1 bar and (**b**) 10 bar of pressure.

**Figure 11 materials-16-00117-f011:**
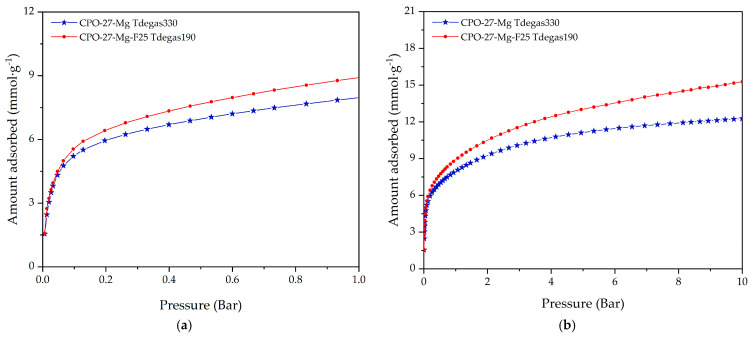
CO_2_ adsorption isotherms at 25 °C for the CPO-27-Mg degassed at 330 °C and CPO-27-Mg-F25 degassed at 190 °C up to (**a**) 1 bar and (**b**) 10 bar of pressure.

**Table 1 materials-16-00117-t001:** Average crystallite size (Dp) of functionalized and non-functionalized MOFs.

Compound	Dp (nm)	% Decrease of Dp
CPO-27-Mg	54.8	-
CPO-27-Mg-F10	40.2	26.59
CPO-27-Mg-F25	40.0	26.95
CPO-27-Mg-F50	50.8	7.33
CPO-27-Mg-F100	45.7	16.55

**Table 2 materials-16-00117-t002:** Textural properties of the CPO-27-Mg compounds.

Sample	*S*_BET_ (m^2^ g^−1^)	*V*_μP-N2_^1^ (cm^3^ g^−1^)	*V*_mP-N2_^2^ (cm^3^ g^−1^)	*V*_TP_^3^ (cm^3^ g^−1^)	*V*_μP-CO2_^1^ (cm^3^ g^−1^)
CPO-27-Mg- Tdegas330	1630	0.63	0.03	0.69	0.60
CPO-27-Mg- Tdegas250	1055	0.40	0.05	0.45	0.40
CPO-27-Mg- Tdegas140	490	0.19	0.05	0.24	0.18

^1^ Calculated with Dubinin–Radushkevich equation at 273 K up to atmospheric pressure. ^2^
*V*_mP-N2_ = *V*_TP_ − *V*_μP-N2_. ^3^ Calculated with Gurvich rule.

**Table 3 materials-16-00117-t003:** CO_2_ adsorption capacities on different types of adsorbents.

Sample	Conditions	CO_2_ Uptake (mmol g^−1^)	Ref.
CPO-27-Mg Tdegas330	1 bar, 298 K	8.01	This work
	10 bar, 298 K	12.26	
CPO-27-Mg-F25 Tdegas190	1 bar, 298 K	9.0	This work
	10 bar, 298 K	15.27	
CPO-27-Mg-F10 Tdegas190	1 bar, 298 K	6.7	This work
	10 bar, 298 K	11.2	
Mg-MOF74	1 bar, 296 K	5.91	[14]
ED-Mg/DOBDC (ethylene diamine functionalized)	400 ppm CO_2_/Ar, 298 K	1.51	[12]
ED-Mg/DOBDC (ethylene diamine functionalized)	1 bar, 298 K	4.66	[13]
CPO-27-Mg-c (ethylene diamine functionalized)	1 bar, 298 K	5.4	[16]
TEPA-Mg-MOF-74 (TEPA functionalized)	1 bar, 298 K	6.11	[17]
Cu_3_(BTC)_2_	1 bar, 315 K	2.5	[36]
Ni_3_(BTC)_2_(Me_2_NH)_2_(H_2_O)	1 bar, 313 K	2.05	[25]
CPC-700 (polymer-derived N-doped carbon)	10 bar, 298 K	14.1	[37]
HPC5b2-1100 (MOF-derived hierarchical carbon)	10 bar, 298 K	13.9	[38]
AS-2-600 (activated carbon)	1 bar, 298 K	4.8	[39]
HB (hydrotalcite—layered double hydroxide)	1 bar, 343 K	1.8	[40]
LiX Zeolite	10 bar, 298 K	8.87	[41]

## Data Availability

Not applicable.

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
