# Peer review of "Strategies for Improving the CO2 Adsorption Process of CPO-27-Mg through Thermal Treatment and Urea Functionalization"

_materials, 2022, doi:10.3390/ma16010117_

Round 1

Reviewer 1 Report

The work of Godoy and coworkers could be interesting to the audience of the journal. However, the quality of presentation is poor – both basic science and flow of presentation. There is also a writing issue, which should be avoided consulting a native English writer.

1-      There is no simulation in this study, so experimental results are not confirmed by theory. Can the authors perform simulation using DFT and revise their paper?

2-      Peaks of FTIR spectra in Figure 2 should be marked with wavenumbers, and thoroughly discussed. Comparison with literature is a must to ensure that the bands are correctly assigned.  

Author Response

Reviewer 1

The work of Godoy and coworkers could be interesting to the audience of the journal. However, the quality of presentation is poor – both basic science and flow of presentation. There is also a writing issue, which should be avoided consulting a native English writer.

We thank the reviewer for his suggestions and comments on our work. Based on this, the writing of the work has been carefully analyzed and modified.

  1. There is no simulation in this study, so experimental results are not confirmed by theory. Can the authors perform simulation using DFT and revise their paper?

In the present work, our objective was to investigate the incidence of the degree of urea functionalization on the CO2 adsorption capacity of CPO-27-Mg and to compare this methodology with thermal treatments varying the degassing temperature of the non-functionalized MOF. Although DFT simulations would be helpful to correlate both experimental and theoretical results by understanding the system more deeply, we could not perform these calculations due to time constraints; nonetheless, this tool will be considered for further work.

  1. Peaks of FTIR spectra in Figure 2 should be marked with wavenumbers, and thoroughly discussed. Comparison with literature is a must to ensure that the bands are correctly assigned. 

We have modified Fig. 2, including the wave numbers of the most important signals of the different compounds. In addition, we have included references that support the correct assignment of the bands, both for CPO-27-Mg and urea.

Reviewer 2 Report

 The paper addresses the important problem of CO2 removal. The authors have clearly presented the assumptions and methodology of the study. However, several points need to be corrected and clarified.
1. The abstract must be rewritten. Writing in the form of "we" is not advisable in this section of manuscript.
2. The authors have carried out several studies on the characterisation of the materials obtained. Whether a correlation was observed between the characteristics analysed and the ability to CO2 remove. Would it be possible to predict from the baseline characteristics whether the material would be suitable for the proposed method?
3. Due to the applied nature of the research, it would be worthwhile to add studies or at least a commentary on the manageability of the material after the process, reusability, and regeneration.
4. In conclusion, the best option, according to the authors, for carrying out the carbon dioxide removal process should be added.

Author Response

Reviewer 2

The paper addresses the important problem of CO2 removal. The authors have clearly presented the assumptions and methodology of the study. However, several points need to be corrected and clarified.

  1. The abstract must be rewritten. Writing in the form of "we" is not advisable in this section of manuscript.

The abstract of the manuscript has been modified as requested.

  1. The authors have carried out several studies on the characterisation of the materials obtained. Whether a correlation was observed between the characteristics analysed and the ability to CO2 remove. Would it be possible to predict from the baseline characteristics whether the material would be suitable for the proposed method?

According to our results, it is expectable that the materials studied in this work present CO2 remotion capabilities in “real life” applications. Nonetheless, our predictions will be limited at the lab scale because to predict the behavior at bigger scales, we need to consider other conditions beyond this work’s scope.

  1. Due to the applied nature of the research, it would be worthwhile to add studies or at least a commentary on the manageability of the material after the process, reusability, and regeneration.

It is a good suggestion; unfortunately, we do not have studies on the reusability or regeneration of the materials. However, to understand these materials’ life cycle, we are currently planning a series of adsorption-desorption processes to evaluate their cyclability. Once these MOFs reach the end of their lifetime, they can be calcined to produce porous carbons, whereas the metal can be recovered and reused.

  1. In conclusion, the best option, according to the authors, for carrying out the carbon dioxide removal process should be added.

According to the reviewer’s observation, the best option was added to the new manuscript.

Reviewer 3 Report

Does Average Crystallite size (Dp) matches TEM  imaging?

Can the author explain why m/z 29 and 31 are characteristic of methanol?

Urea decomposite into amine and ammonia and isocyanic acid. Does the author see any MS signal at 43?

Figure 7, should be “activation temperature” or “degas temperature” for the axes of the lower figure.

Author Response

Reviewer 3

  1. Does Average Crystallite size (Dp) matches TEM imaging?

The determination of the Dp for the different samples was carried out with the aim of using the XRD data to analyze the incidence of the functionalization treatment on the crystallinity of the starting material. Unfortunately, within the set of characterization techniques that we implement for the study of materials, we do not perform TEM microscopy.

  1. Can the author explain why m/z 29 and 31 are characteristic of methanol?

As can be seen in its mass spectrum, the fragments m/z 29 and 31 are, in addition to the molecular ion m/z 32, the signals with the highest intensity for methanol. The probable fragmentation mechanism and the related species for both fragments are presented below:

  1. Urea decomposite into amine and ammonia and isocyanic acid. Does the author see any MS signal at 43?

As seen in the urea MS, we have evaluated only the most important signals that arise from its rupture, i.e., m/z 17 and 44. The fragment m/z 43, corresponding to isocyanic acid, is unstable, so it does not have appreciable intensity in the MS.

We add a pdf file a figure with these molecules and the corresponding m/z relation

  1. Figure 7, should be “activation temperature” or “degas temperature” for the axes of the lower figure.

We appreciate the reviewer's suggestion. The x-axis in Fig. 7 has been modified to "degas temperature."

Round 2

Reviewer 1 Report

Authors replied to my first concern as: 

In the present work, our objective was to investigate the incidence of the degree of urea functionalization on the CO2 adsorption capacity of CPO-27-Mg and to compare this methodology with thermal treatments varying the degassing temperature of the non-functionalized MOF. Although DFT simulations would be helpful to correlate both experimental and theoretical results by understanding the system more deeply, we could not perform these calculations due to time constraints; nonetheless, this tool will be considered for further work.

Comment: Authors have replied that they are running out of time. So, they do not have time to provide additional data to support their experimental findings. I found this attitude of the authors very surprising. Since authors have failed to provide additional data because of their time constraints, the data presented in this paper is deemed unsuitable for publication. And I do not recommend publication of this work. 

Author Response

We think we owe an apology to the reviewer since, in reality, we do not have anyone in our group who does DFT calculations at the moment, and requesting them from other work teams would take us a lot of time.